# A Structural Design Method of 3D Electromagnetic Wave-Absorbing Woven Fabrics

**DOI:** 10.3390/polym14132635

**Published:** 2022-06-28

**Authors:** Jianjun Yin, Wensuo Ma, Zuobin Gao, Xianqing Lei, Chenhui Jia

**Affiliations:** School of Mechatronic Engineering, Henan University of Science and Technology, 48 Xiyuan Road, Luoyang 471003, China; jianjun1010@126.com (J.Y.); gaozuobin@haust.edu.cn (Z.G.); lxq@haust.edu.cn (X.L.); 9903170@haust.edu.cn (C.J.)

**Keywords:** structural design method, 3D woven fabric, wave-absorbing fabric, electromagnetic parameters, glass fibers, carbon fibers

## Abstract

Based on the wave absorption model of 3D woven fabric and the zero-reflection equations, a new structural design method of 3D electromagnetic (EM) wave-absorbing woven fabrics was obtained. The 3D woven fabrics fabricated by the proposed method had the structure of a bidirectional angle interlock. Continuous S-2 glass fibers were used as the matching layer of this 3D woven fabric, and continuous carbon fibers were used as the absorbing layer. The absorbing layer satisfied the equivalent EM parameters under the condition of zero reflection. The results of the simulation and experiment showed that the performance trends of the 3D wave-absorbing fabric obtained by this method were consistent with the theory, which verified the correctness of the structure design method. The 3D fabrics obtained by this method have the advantages of wide absorbing frequencies and good absorbing performance (−20 dB). This structural design method also has theoretical guiding significance for the development of 3D wave-absorbing fabrics.

## 1. Introduction

The application of EM waves has been exploded with the development of electronic equipment and communication technology [1]. In order to effectively protect and control EM waves, the requirements for wave-absorbing materials from all walks of life are more demanding than ever before [2]. New kinds of wave-absorbing materials with good absorption performance, wide working frequency, thin width and lightweight properties are the research goal of many scholars [3,4,5]. Three-dimensional woven fabrics have been widely used in wave-absorbing materials due to their characteristics of strong designability, high specific strength, high specific modulus, high damage tolerance and the ability to form complex structures at the same time [6,7].

According to the different positions of the absorbing body, the 3D wave-absorbing fabrics can be divided into coated types and structure types [8]. The coated type of absorbing fabric applies wave-absorbing agents such as carbon black, graphite, ferrite and other resistive or magnetic medium materials to the inside or on the surface of the fabric. The 3D woven fabric is used as a stable support structure, and the absorbing agent plays a role in absorbing EM waves [9,10]. Xie et al. used carbon black (CB) as the wave absorbent and embedded it into 3D woven composites. The results showed that the absorbing properties of 3D woven composites were improved obviously. Three-dimensional woven composites can reduce the complex dielectric constant of the composite, which can improve the impedance matching of the composite, reducing the reflection of EM waves and improving the absorbing properties [11]. Simayee et al. mixed micromagnetic carbonyl iron powder with nano-carbon black as the wave absorbent and coated it on polyester fabrics by filling–drying curing and aluminum sputtering coating. The experiment showed that the aluminized polyester fabrics coated with carbonyl iron powder and nano-carbon black had better absorption [12]. Liu et al. used ferrite and silicon carbide as wave absorbents at the bottom and surface of woven polyester fabrics. The ferrite/SiC double-coated polyester fabric material with absorbing properties was prepared by optimizing the EM parameters. The results showed that it had the best absorbing performance at the frequency of 10 GHz [13].

Structure types of wave-absorbing fabrics are made of yarns or fiber bundles with absorbing properties, such as nickel–iron or carbon fiber, which are directly woven or woven into the finished fabric based on a stable structure. The absorbing properties and influencing parameters of 3D fabrics have been tested by experiments [14,15]. Fan Wei et al. tested three kinds of 3D woven carbon fiber/epoxy composites with different structures, and the experiments showed that the composites had good EM absorption and shielding efficiency, but also excellent mechanical properties and absorption capacity, and they can be widely used in radar absorption structures [16]. Then they tested the mechanical properties and absorbing properties of 3D orthogonal fabric. The results showed that compared with the layered composite structure, it had better mechanical properties and better absorbing properties for radar waves [17]. Ayan et al. conducted experimental tests on cotton fabrics, carbon fabrics and a cotton–carbon fabric composite board with 3–18 GHz frequencies. The mechanical value of the cotton fabric composite board was lower, but the wave absorption value in a certain frequency range was higher than the carbon fabric composite board. The absorbing performance of the cotton–carbon fabric composites was better than that of pure carbon composites at frequencies of 12–18 GHz [18].

For coated types of wave-absorbing fabrics, although the research on their absorbance is relatively mature, there are many problems, such as a narrow absorbance frequency and single absorbance performance, as it is hard to have both a wide absorbing frequency and good absorbing performance. The research into structure types of wave-absorbing fabrics has concentrated on the type of wave-absorbing fiber but not the fabric structure. However, there are few reports on the effect of fabric structure on the absorbing properties, and different fabric structures have different effects on the absorbing properties. It is of guiding significance for the development of absorbing materials to study the effect of the fabric structure on the absorbing properties and put forward an optimization design method for 3D wave-absorbing fabric structures.

## 2. The Structural Design Method

### 2.1. The Equivalent EM Parameter Model of 3D Woven Fabrics

In previous studies, an equivalent EM parameter model of 3D woven fabrics was presented, based on the strong fluctuation theory [19]. According to this model, the equivalent EM parameters of 3D woven fabrics with different structures can be obtained from continuous fibers, as shown in Equation (1).
(1)$$\left\{ \begin{array}{l} \varepsilon_{⊥}=\varepsilon_{a}V_{f}-V_{f}-\frac{\varepsilon_{a}}{2}+\frac{\sqrt{4\varepsilon_{a}{}^{2}V_{f}{}^{2}-4\varepsilon_{a}{}^{2}V_{f}+\varepsilon_{a}{}^{2}-8\varepsilon_{a}V_{f}{}^{2}+8\varepsilon_{a}V_{f}+2\varepsilon_{a}+4V_{f}{}^{2}+1}}{2}+\frac{1}{2}\\ \mu_{⊥}=\mu_{a}V_{f}-V_{f}-\frac{\mu_{a}}{2}+\frac{\sqrt{4\mu_{a}{}^{2}V_{f}{}^{2}-4\mu_{a}{}^{2}V_{f}+\mu_{a}{}^{2}-8\mu_{a}V_{f}{}^{2}+8\mu_{a}V_{f}+2\mu_{a}+4V_{f}{}^{2}+1}}{2}+\frac{1}{2} \end{array}\right.$$
where $\varepsilon_{⊥}$ and $\mu_{⊥}$ are the equivalent dielectric constant and the permeability perpendicular to the direction of incident EM waves, respectively, and $\varepsilon_{a}$ and $\mu_{a}$ are the equivalent dielectric constant and the permeability of continuous fibers, respectively. $V_{f}$ is the fiber content percentage of the 3D woven fabrics, as shown in Equation (2).
(2)$$\left\{ \begin{array}{l} V_{f}=U_{f}/U\times 100\%\\ U_{f}=4\pi abl\\ U=L^{2}H \end{array}\right.$$
where $U_{f}$ is the volume of a unit of yarn; a and b are the major and minor axes of the ellipse, respectively; l is the value of the crankshaft yarn; and L and H are the height and width of the representative volume unit, respectively, as shown in Figure 1 and Figure 2.

As can be seen from Equations (1) and (2), if the equivalent EM parameters of 3D woven fabrics are given, the fiber content percentage of the fabrics can be calculated, and thus the structural parameters of the fabrics can be obtained and a 3D woven fabric structure design that meets different EM requirements can be realized.

### 2.2. The Equations Satisfying the Condition of Zero Reflection

The ideal situation of wave-absorbing fabric is to achieve zero reflection of electromagnetic waves. It can be obtained from transmission line theory that [20,21]:(3)$$\{\begin{array}{l} Z_{in}=\sqrt{\frac{\mu }{\varepsilon }}\tanh(\gamma d)\\ Z_{0}=\sqrt{\frac{\mu_{0}}{\varepsilon_{0}}}\\ \gamma =\alpha +j\beta =j\frac{2\pi }{\lambda }\sqrt{\mu \varepsilon }\\ \varepsilon =\varepsilon^{\prime }-j\varepsilon^{\prime \prime }=\varepsilon^{\prime }(1-j\tan\delta_{\varepsilon })\\ \mu =\mu^{\prime }-j\mu^{\prime \prime }=\mu^{\prime }(1-j\tan\delta_{\mu }) \end{array}$$
where $Z_{in}$ and $Z_{0}=120\pi (\Omega )$ are the input impedance and the free space impedance, respectively; γ is the propagation constant; *d* is the thickness of the material; λ is the wavelength; μ, $\mu^{\prime }$, $\mu^{\prime \prime }$ and $\tan\delta_{\mu }$ are the complex permeability, the real permeability and the imaginary permeability of the material, respectively; ε, $\varepsilon^{\prime }$, $\varepsilon^{\prime \prime }$ and $\tan\delta_{\varepsilon }$ are the complex dielectric constant, the real part of the dielectric constant, the imaginary part of the dielectric constant and the tangent of the electric loss angle of the material, respectively; α is the attenuation constant of the EM wave in a medium; β is the phase coefficient; and $\mu_{0}=4\pi \times 10^{-7}(H/m)$ and $\varepsilon_{0}=8.854\times 10^{-12}(F/m)$ are the permeability and dielectric constant in free space, respectively.

It can be obtained from Equation (3) that:(4)$$\left\{ \begin{array}{l} \alpha =\frac{\pi }{\lambda }\sqrt{2\mu^{\prime }\varepsilon^{\prime }}\sqrt{\tan\delta_{\varepsilon }\tan\delta_{\mu }-1+\sqrt{\tan^{2}\delta_{\varepsilon }\tan2\delta_{\mu }+\tan^{2}\delta_{\varepsilon }+\tan^{2}\delta_{\mu }+1}}\\ \beta =\frac{2\pi }{\lambda } \end{array}\right.$$

The condition of zero reflection between far-field EM waves and a uniform dielectric composite surface is that the material’s impedance matches the input free-space impedance, which means $\frac{Z_{in}}{Z_{0}}=1$, and it can be obtained that:(5)$$\sqrt{\frac{\mu /\mu_{0}}{\varepsilon /\varepsilon_{0}}}\tanh(\gamma d)=1$$
where:(6)$$\sqrt{\frac{\mu }{\varepsilon }}=\sqrt{\frac{\mu^{\prime }(1-j\tan\delta_{\mu })}{\varepsilon^{\prime }(1-j\tan\delta_{\varepsilon })}}=\sqrt{\frac{\mu^{\prime }\cos\delta_{\varepsilon }}{\varepsilon^{\prime }\cos\delta_{\mu }}}\left[\cos\frac{\left(\delta_{\mu }-\delta_{\varepsilon }\right)}{2}-j\sin\frac{\left(\delta_{\mu }-\delta_{\varepsilon }\right)}{2}\right]$$
(7)$$\frac{1}{\tanh(\gamma d)}=\frac{\sinh(2\alpha d)}{\cosh(2\alpha d)-\cosh(2\beta d)}-j\frac{\sinh(2\beta d)}{\cosh(2\alpha d)-\cosh(2\beta d)}$$

From Equations (5)–(7), it can be obtained that:(8)$$\left\{ \begin{array}{l} \frac{\mu^{\prime }\cos\delta_{\varepsilon }}{\varepsilon^{\prime }\cos\delta_{\mu }}=\frac{\cosh(2\alpha d)+\cos(2\beta d)}{\left[\cosh(2\alpha d)-\cos(2\beta d)\right]}\\ \tan\frac{\delta_{\mu }-\delta_{\varepsilon }}{2}=\frac{\sin(2\beta d)}{\sinh(2\beta d)} \end{array}\right.$$

Equation (8) is the equation satisfying the zero surface reflection of a vertically incident plane wave to one single uniform layer of an absorbing material with a thickness *d.* As can be seen from the equation, if the EM parameters of the absorbing material are known, the thickness required to achieve zero reflection can be calculated. Similarly, if the thickness is given, the EM parameters of the material can be approximated.

### 2.3. Calculation of the Equivalent EM Parameters of a Single Layer under Initial Conditions

By solving Equation (8) in Matlab, it can be obtained that:(9)$$\left\{ \begin{array}{l} 2\beta d=\frac{\pi }{1+\tan{\left(\frac{\delta_{\mu }+\delta_{\varepsilon }}{2}\right)}^{2}}\begin{array}{cc} & \left(\tan(\frac{\delta_{\mu }+\delta_{\varepsilon }}{2})<0.25\right) \end{array}\\ 2\beta d=\sqrt{\frac{6(1-\tan{\left(\frac{\delta_{\mu }+\delta_{\varepsilon }}{2}\right)}^{2})}{1+\tan{\left(\frac{\delta_{\mu }+\delta_{\varepsilon }}{2}\right)}^{2}}}\begin{array}{cc} & \left(\tan(\frac{\delta_{\mu }+\delta_{\varepsilon }}{2})>0.25\right) \end{array} \end{array}\right.$$
where $\delta_{\mu }=0$; $\tan(\frac{\delta_{\mu }+\delta_{\varepsilon }}{2})=\alpha /\beta$.

We assumed that the thickness of the absorbing fabric was required to be 1 cm and braided from continuous carbon fibers with a steady-state complex dielectric constant value of 4922-535i [22]. For the convenience of calculation, it was assumed that the loss caused by the carbon fiber was only the dielectric loss, so the relative permeability of carbon fiber is 1 and $\delta_{\mu }=0$. We set three initial conditions: S1, S2 and S3, and let the incident frequencies be 5, 8 and 12 GHz. These were substituted into Equation (9) to obtain the equivalent EM parameters satisfying zero reflection, as shown in Table 1. Next, Equation (2) was used to design fabric parameters that met the value of $V_{f}$ in Table 1.

Reflection loss obtained from the literature [23].
(10)$$\left\{ \begin{array}{l} R(dB)=20\log\left|\frac{Z_{in}-1}{Z_{in}+1}\right|\\ Z_{in}=\sqrt{\frac{\mu_{g}}{\varepsilon_{g}}}\tanh\left[j\frac{2\pi fd}{c}\sqrt{\mu_{g}\varepsilon_{g}}\right]\\ Z_{0}=\sqrt{\frac{\mu_{0}}{\varepsilon_{0}}} \end{array}\right.$$

By substituting the initial conditions and the EM parameters in Table 1 into Equation (10), we obtained the following.

Figure 3a–c is the schematic diagram of reflection loss under equivalent electromagnetic parameters at the given frequencies of 5, 8 and 12 GHz. In Figure 3a, the absorption peak of the front part was between 4 and 7 GHz. The absorption peak of Figure 3b was at 8 GHz. The absorption peak of Figure 3c was between 10 and 12 GHz. The absorption peaks of the region corresponded to the given frequencies, and the reflection loss in the given frequencies range was less than −10 dB (the absorption rate was greater than 90%). It can be shown that this method can effectively design a fabric structure which can achieve the absorbing performance at a specified frequency. However, a single layer of the wave-absorbing fabric has a narrow peak frequency bandwidth, which is not conducive to the development of fabrics absorbing wide frequencies.

### 2.4. A Structural Design for Multilayer Absorbing Fabrics

The method of calculating the equivalent EM parameters based on the strong fluctuation theory is equivalent for a multilayer heterogeneous fabric and a single-layer homogeneous fabric [24], but fabric with single layer structure is unable to achieve both high absorption and great impedance matching, so the absorption peak is narrow. In order to meet the requirements of absorbing performance under a wide frequency band, a multilayer structure could be designed. Impedance matching means the EM waves can enter into the fabric to the maximum extent. The outermost layer is the wave transmitting layer; the second layer is the wave-absorbing layer, and the third layer is a highly reflective layer with a metal substrate, as shown in Figure 4.

The reflection loss of multilayer media can be obtained from [25,26]:(11)$$\left\{ \begin{array}{l} R(dB)=20\log\left|\frac{Z_{inN}-1}{Z_{inN}+1}\right|\\ Z_{ink}=Z_{k}\frac{Z_{ink-1}+Z_{k}\tanh(\gamma_{k}d_{k})}{Z_{k}+Z_{ink-1}\tanh(\gamma_{k}d_{k})}\\ Z_{k}=\sqrt{\frac{\mu_{gk}}{\varepsilon_{gk}}} \end{array}\right.$$
where $Z_{ink}$ represents the input impedance of layer *k*; $Z_{k}$ represents the impedance of layer *k*; $\mu_{gk}$ and $\varepsilon_{gk}$ represent the complex permeability and dielectric constant of layer *k*, respectively; and $\gamma_{k}$ and $d_{k}$ represent the propagation constant and thickness of layer *k*, respectively.

S-2 glass fiber has a low dielectric constant and good mechanical properties, as shown in Table 2, and the matching layer woven from S-2 continuous glass fiber has a relative dielectric constant of 5.2 [27]. The equivalent EM parameters of the absorbing layer obtained in Table 1 were substituted into Table 3.

Substituted Table 3 into Equation (11), as shown in Figure 5:

The effective absorption performance (<−10 dB) of P1 was in the frequency range of 11–18 GHz. P2 had an effective absorbing performance of 13–18 GHz. The effective absorption performance of P3 was 14–18 GHz. The higher the frequencies, the better the absorption performance. The peak value of absorption increased with an increase in the frequency. This was because the absorption mechanism of carbon fibers belongs to dielectric loss, and under the action of EM waves, the dielectric material was polarized repeatedly, leading to rotation and orientation, and generating dielectric loss. At the same time, when the frequencies of the external electric field are consistent with the thermal vibration frequencies of the molecule or atom, it will cause resonance loss, which leads to enhancement of the wave-absorbing capacity with an increase in the EM frequencies. The absorption performance of P1 is greater than that of P2 and P3, indicating that the sample with a larger dielectric constant and a larger fiber volume content had better absorption performance. Compared with single-layer absorbing fabrics (Figure 3), multilayer absorbing fabrics had wider effective absorbing frequencies and better absorbing performance at larger EM frequencies. First, the wave-absorbing layer can be designed by the method of Section 2.3, and then the matching layer and a metal substrate can be added, and finally, the multilayer wave-absorbing woven fabrics will be obtained.

## 3. The Simulation

CST (Computer Simulation Technology) is one of the most widely used finite element EM wave simulation software tools with high accuracy. A representative volume unit model of the 3D woven fabric was imported into the software. The boundary condition was set as periodic, and the frequency range of the incident EM waves from the top of the fabric were 1~18 GHz, as shown in Figure 6.

We used glass fibers as the matching layer and carbon fibers as the absorbing layer. By substituting the parameters of P1, P2 and P3 in Table 2 into the model, the following results can be shown in Figure 7:

Figure 8a−c shows a comparison between the theoretical values and the simulation of samples P1−P3 respectively. The effective absorption frequencies of P1, P2 and P3 were 13−18 GHz, 14−18 GHz and 14−18 GHz, respectively. The overall variation trend was consistent with the theoretical values, and the simulation was close to the theory in the frequencies of 10−18 GHz. The simulation values were slightly larger than the theoretical values, and the absorbing performance increased with an increase in the frequency. The absorption performance was ranked P3 > P2 > P1: the higher the fiber content, the better the absorption performance. These trends were consistent with the theoretical values, which proved the correctness of the theory.

## 4. The Experiment

### 4.1. Preparation of the Samples

Equation (2) was used to write a program to obtain the variation trend of the fiber volume fraction with the yarn deformation factor shown in Figure 9, where k=a/b.

We designed the fabric’s structural parameters according to the fiber volume fraction in Figure 9. The weaving process of the 3D bidirectional angle interlock woven fabric is shown in Figure 10 [6]. Three samples were woven from continuous carbon fibers as shown in Figure 11, and the samples’ parameters are shown in Table 4 and Table 5.

Figure 11a,b shows top view and side view of the 3D bidirectional angle interlock woven carbon fiber fabric; Figure 11c shows sample P1 with S2-glass fibers as the matching layer, as shown in Table 2.

### 4.2. Experimental Methods

This experiment used the free space method, which adopted the opening EM characteristic parameter testing technology first introduced by Cullen, who deduced the theory of the EM parameters of materials measured in free space [28]. This method is a non-destructive and non-contact method, and only a pair of double-sided parallel plates was required, so the number of samples required was low. The ZNB40 vector network analyzer was applied in this experiment, as shown in Figure 12. Based on GJB2038-1994 standards, the free space test process was as shown in Figure 13.

First, the ZNB40 vector network analyzer generated a sweep signal. Next, after being amplified by a power amplifier, this signal was transmitted as an EM wave by a transmitting antenna. The EM wave passed through the samples and was received by a receiving antenna, which converted it into a reflected signal and returned it to the ZNB40 vector network analyzer; thus, reflection parameters and transmission parameters could be obtained. The reflection loss, absorption loss and transmission loss of the EM wave were calculated according to the amplitude and phase of the S scattering parameters. The EM wave was vertically incident to the sample surface. The right side of the samples was a metal plate substrate. The two antennae (transmitting and receiving antennae) had a wide band frequency range of 1–18 GHz and the direction of electric field was parallel to the weft yarn’s braiding direction. When the EM waves passed through the material, this produced reflection loss (R), absorption loss (A), multiple reflection loss (M) and transmission loss (T). There was a metal substrate in this sample, so the transmission loss was 0. As the samples were wave-absorbing materials, multiple reflection loss could be ignored, and it was obtained that:(12)T=1−R−A

The signals received by Port 1 ($S_{11}$) and Port 2 ($S_{21}$) were the reflection loss and transmission loss, respectively. Equation (13) showed that:(13)$$R={\left|\frac{E_{R}}{E_{i}}\right|}^{2}={\left|S_{11}\right|}^{2}$$
where $E_{R}$ is the induced voltage of the reflected EM wave and $E_{i}$ is the induced voltage of the incident EM wave. In decibel form, it can be obtained as:(14)$$R_{L}=20\mathrm{lg}\left|S_{11}\right|(dB)$$

### 4.3. Results and Discussion

Each sample (P1–P3) was measured five times and averaged, as shown in Figure 14. The 95% confidence intervals of P1 were (−4.63, −13.78), those of P2 were (−4.25, −12.93) and those of P3 were (−4.09, −11.48).

Figure 15a–c shows the comparison between the theoretical and experimental values for sample P1–P3, respectively. The experimental results showed that the absorption performance increased with an increase in the frequency, and the absorption performance was ranked P1 > P2 > P3 in the range of 11–18 GHz, which showed that the higher the fiber volume fraction, the stronger the absorption capacity. The experimental values were steeper than the theoretical values, and the absorption capacity was slightly larger than the theoretical value. Because the sample’s surface was not as smooth as the theoretical fabric, thus the actual dielectric constant was higher than the theoretical value, resulting in a slightly better absorption performance. Besides, the actual relative permeability of the sample was not 1. The overall trend of the experimental value was consistent with the theoretical value, which proves the correctness of the theory. When the frequencies exceeded 17 GHz, the deviation between the experimental values and the theoretical values became larger. The reason is that the carbon fiber in the absorbing layer has a certain conductive performance, and due to an increase in the frequency, the skin effect occurred on the sample’s surface and an induced current was generated, so the reflection effect of the sample increased and the absorption loss decreased. However, the theory does not consider the skin effect, and with an increase in the wave frequency, the absorption loss still increases, resulting in a crossover with the experimental results.

## 5. Conclusions

Based on a predicting model for the wave absorption of 3D woven absorbing fabric, the absorbing fabric equation satisfying zero reflection was deduced and a design method of 3D woven absorbing fabric was proposed. This 3D woven fabric with a bidirectional angle interlock structure used S-2 glass fibers as the matching layer and carbon fiber as the absorbing layer. The 3D woven absorbing fabric obtained by this method has the advantages of wider absorbing frequencies and better absorption performance. First, the equivalent EM parameters were calculated by the zero reflection equation under the given initial conditions, then by optimizing the fabric’s structure parameters, a 3D woven fabric satisfying the equivalent EM parameters was obtained. Simulations and measurement experiments verified the correctness of this method, and the higher the fiber volume content fraction of the absorbing fabric, the better the absorbing performance. This method is of guiding significance for the optimal design of absorbing materials.

## Figures and Tables

**Figure 1 polymers-14-02635-f001:**
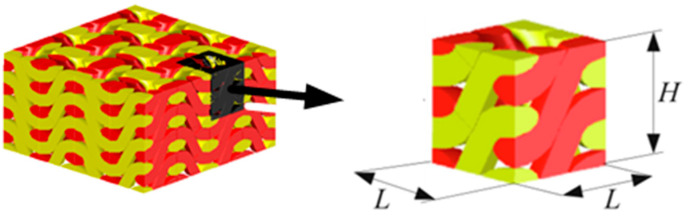
Schematic diagram of 3D woven fabrics.

**Figure 2 polymers-14-02635-f002:**
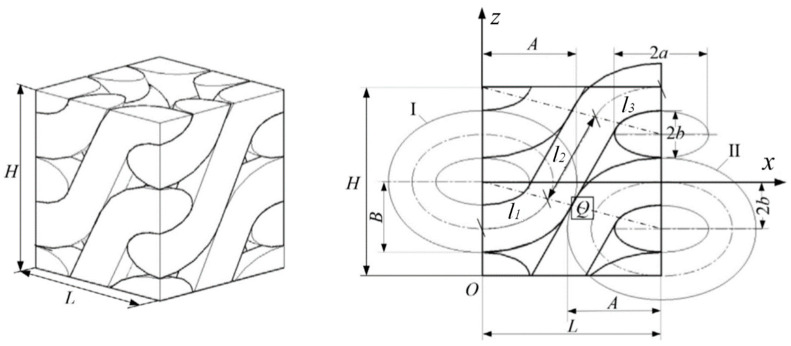
Representative volume unit.

**Figure 3 polymers-14-02635-f003:**
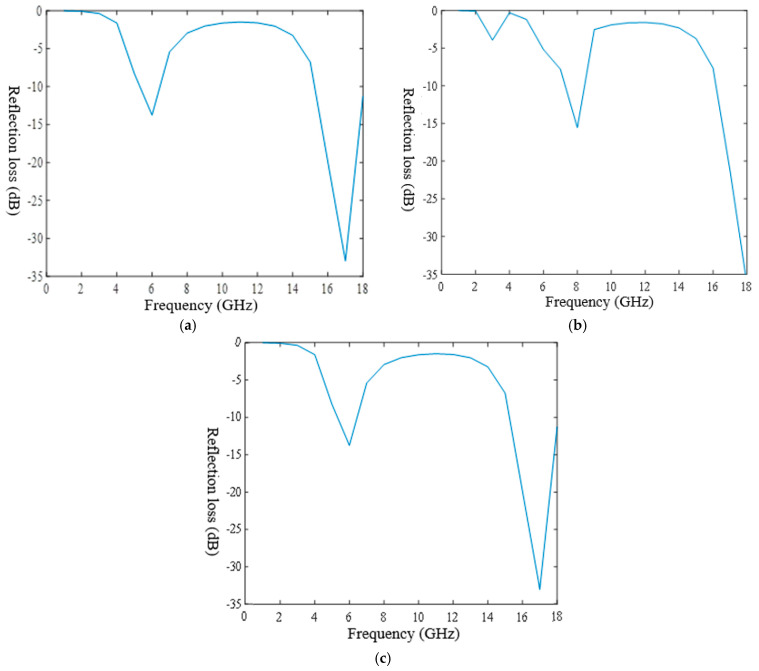
Schematic diagrams of reflection loss under equivalent EM parameters. (**a**–**c**) is the schematic diagram of reflection loss under equivalent EM parameters at the given frequencies of 5, 8 and 12 GHz.

**Figure 4 polymers-14-02635-f004:**
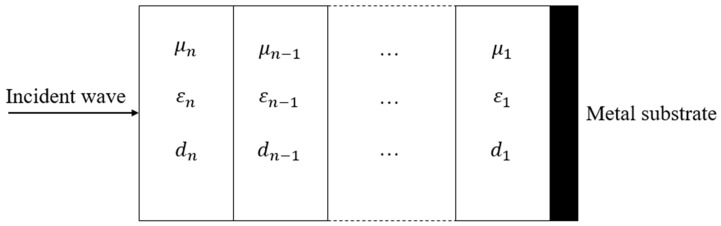
The multilayer wave-absorbing structure.

**Figure 5 polymers-14-02635-f005:**
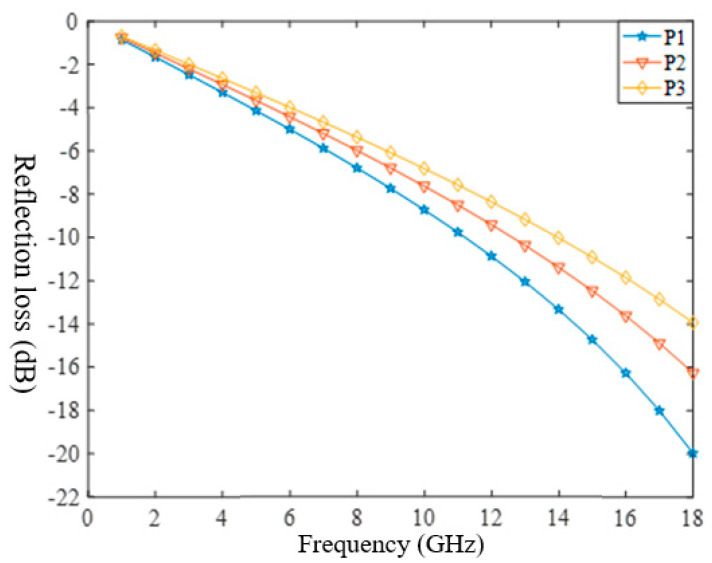
Theoretical values of samples P1–P3.

**Figure 6 polymers-14-02635-f006:**
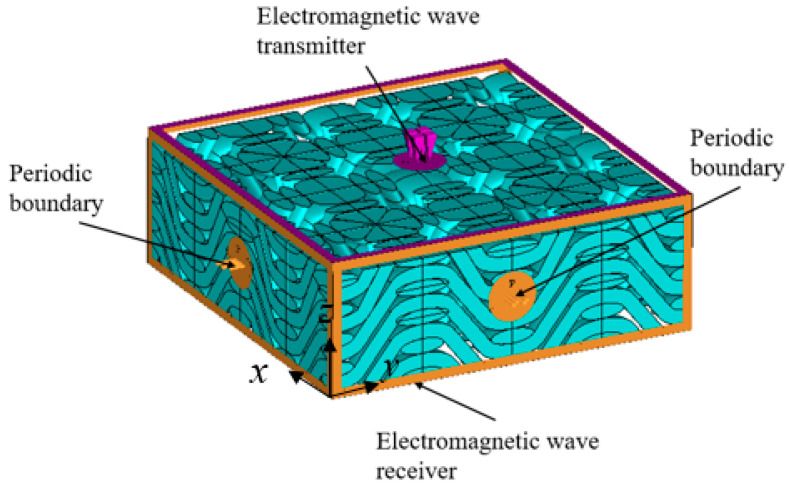
Boundary conditions of the simulation.

**Figure 7 polymers-14-02635-f007:**
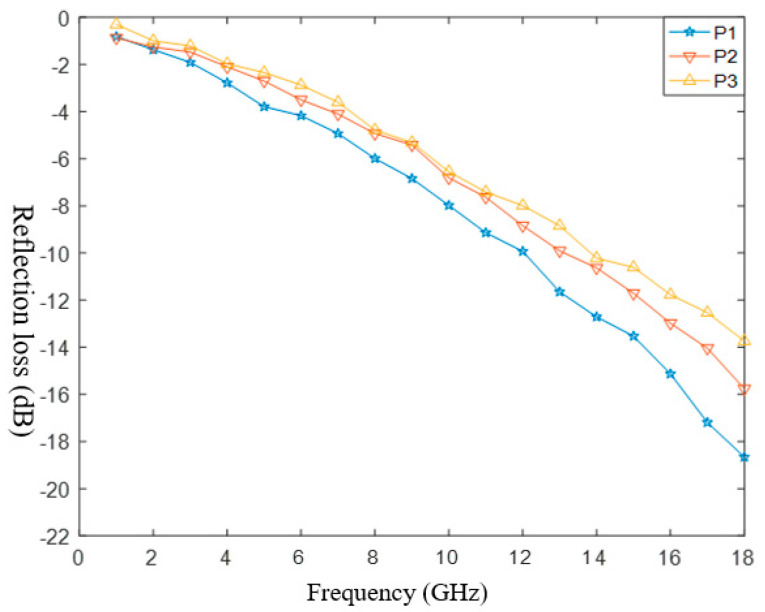
The simulation results.

**Figure 8 polymers-14-02635-f008:**
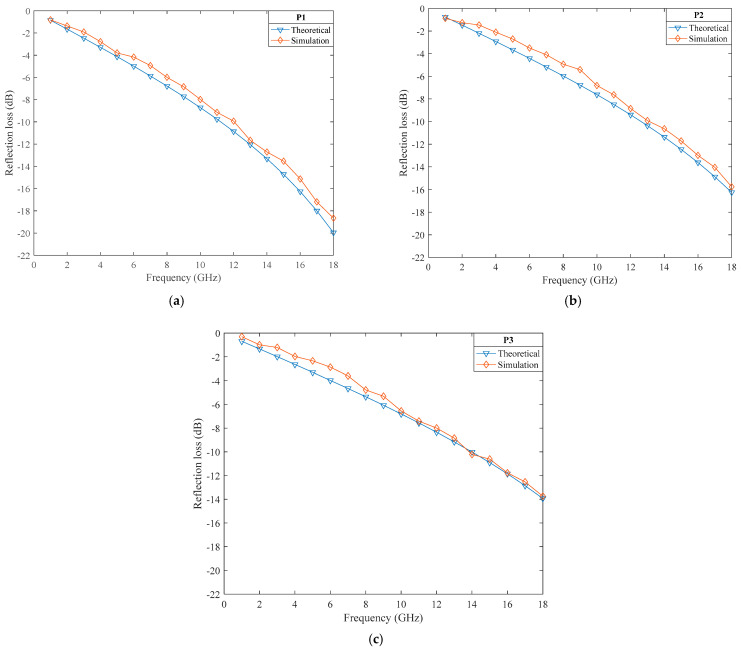
Comparison of the simulation’s results and the theoretical results. (**a**–**c**) shows a comparison between the theoretical values and the simulation of samples P1–P3 respectively.

**Figure 9 polymers-14-02635-f009:**
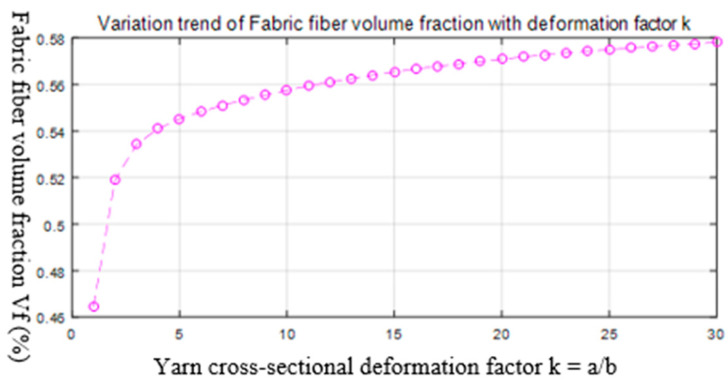
The variation trend of the fiber volume fraction with the yarn deformation factor.

**Figure 10 polymers-14-02635-f010:**
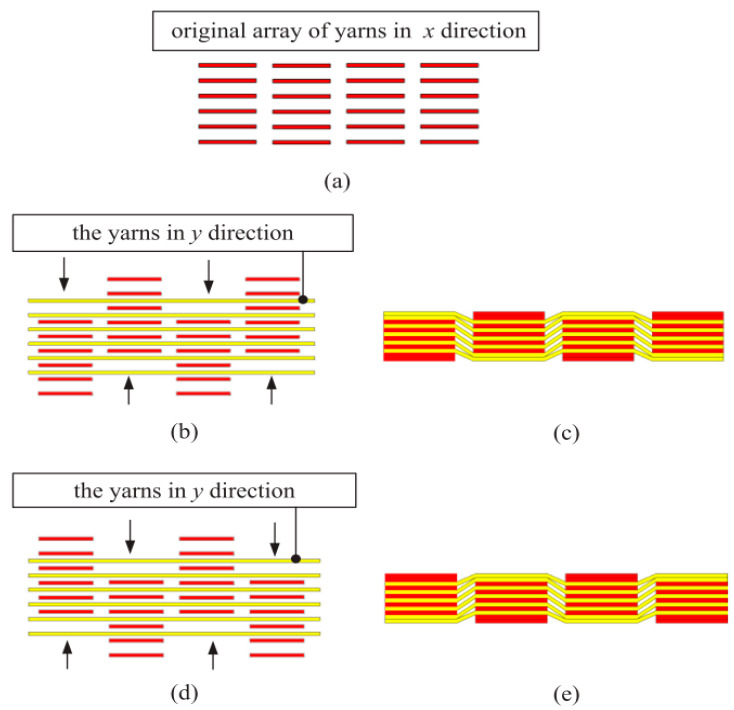
The weaving process of the 3D bidirectional angle interlock woven fabric. (**a**–**e**) are the processing steps.

**Figure 11 polymers-14-02635-f011:**
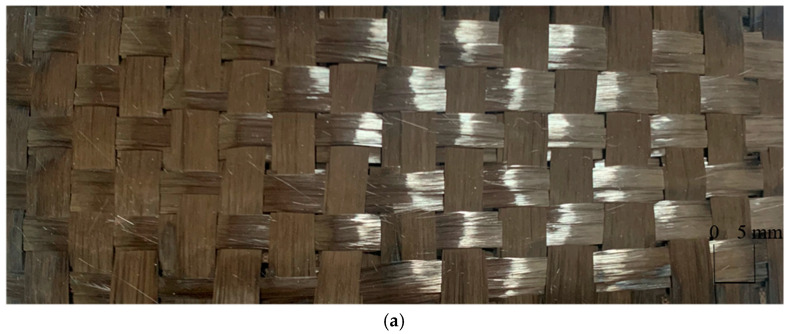
Photo of sample P1. (**a**,**b**) shows top view and side view and (**c**) shows sample P1 with S2-glass fibers as the matching layer.

**Figure 12 polymers-14-02635-f012:**
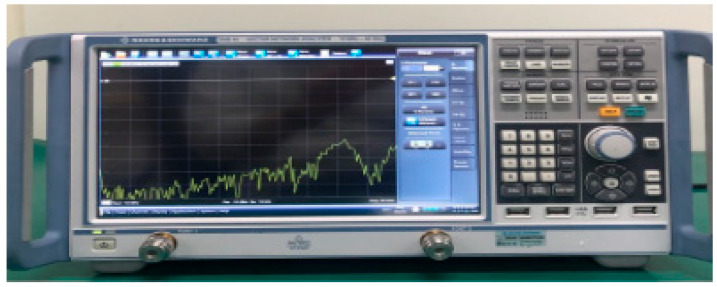
ZNB40 vector network analyzer.

**Figure 13 polymers-14-02635-f013:**
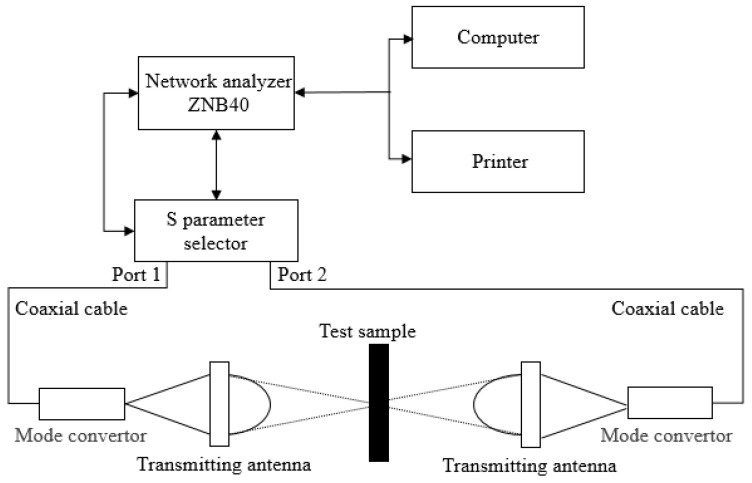
The measurement process.

**Figure 14 polymers-14-02635-f014:**
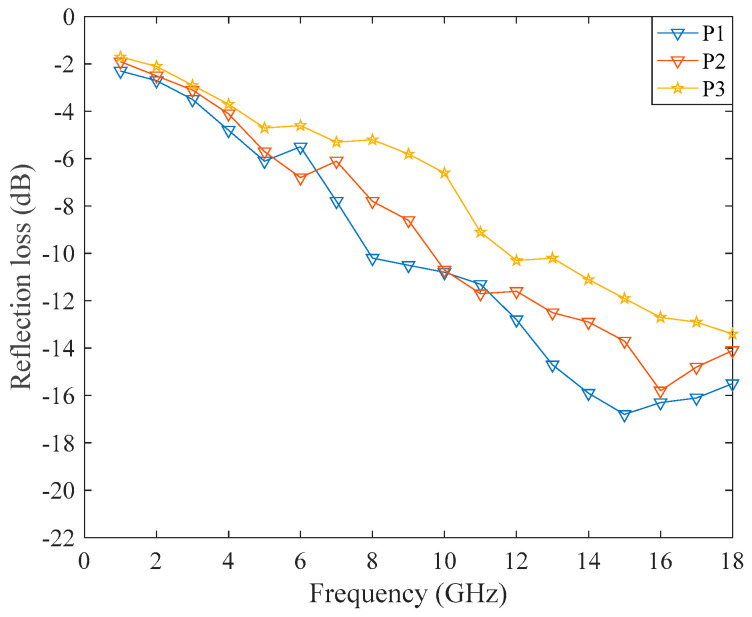
The experimental results.

**Figure 15 polymers-14-02635-f015:**
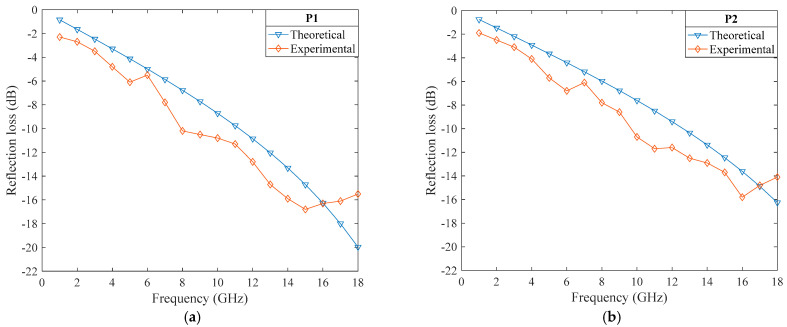
Comparison of the theoretical values and the experimental values. (**a**–**c**) shows the comparison between the theoretical and experimental values for sample P1–P3.

**Table 1 polymers-14-02635-t001:** Samples parameters satisfying zero reflection.

Samples	f (GHz)	$\varepsilon^{\prime }$	$\varepsilon^{\prime \prime }$	$V_{f}$
S1	5	20.2859	220.51	0.4777
S2	8	18.0316	196.00	0.4689
S3	12	16.2868	177.04	0.4605

**Table 2 polymers-14-02635-t002:** Parameters of S-2 glass fibers.

Producer	Form	Tex	Density	Thickness
AGY company	Yarn	66	2.49 g/cm^3^	1 mm

**Table 3 polymers-14-02635-t003:** Equivalent EM parameters from Table 1 derived by substitution into Equation (11).

Sample	d1 (mm)	d2 (mm)	$\varepsilon_{1}$	$\varepsilon_{2}$	$V_{f}$
P1	10	3	20.2859–220.51j	5.2	0.4777
P2	10	3	18.0316–196.00j	5.2	0.4689
P3	10	3	16.2868–177.04j	5.2	0.4605

**Table 4 polymers-14-02635-t004:** Parameters of the carbon fibers.

Producer	Tensile Strength	Tensile Modulus	Density	Form
Toray T800	5880 MPa	294 GPa	1.80 g/cm^3^	No twist

**Table 5 polymers-14-02635-t005:** Parameters of the absorbing layer.

Number of Samples	Types of Fibers	Size (mm × mm × mm)	Parameters of Structure (mm)	Fiber Volume Fraction
P1	T800-16K	150 × 50 × 10	a = 3, b = 1.2, l = 6	0.4777
P2	T800-16K	150 × 50 × 10	a = 3, b = 1.3, l = 6.2	0.4689
P3	T800-16K	150 × 50 × 10	a = 3, b = 1.4, l = 6.4	0.4605

## Data Availability

The data presented in this study are openly available in Web of Science at https://www.sciencedirect.com/science/article/abs/pii/S026382231732038X?via%3Dihub (accessed on 16 May 2022), reference [27].

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
