# Peer review of "A Structural Design Method of 3D Electromagnetic Wave-Absorbing Woven Fabrics"

_polymers, 2022, doi:10.3390/polym14132635_

Round 1

Reviewer 1 Report

The goal of this research is to develop a structural design method of 3D woven absorbing fabric based on the wave absorption model of 3D woven fabric and the zero-reflection equations. The topic is important and critical for 3D wave-absorbing fabrics in terms of EM waves. According to mathematical basis, a well-established design method allows developers to make necessary adjustment to determine the best parameter settings prior to the manufacturing process. In reviewer’s opinion, the experiments are properly conducted and reasonable results are presented. Also, the simulation results of the developed model shows a good agreement with the experimental results.

Author Response

Thank you for your kind reviews.

Reviewer 2 Report

It is difficult to understand (1) the advantages of the texture patterns comparing with conventional absorbers with plate shape or emboss patterns and (2) the real application of electromagnetic absorber of -10 dB at GHz range.

Author Response

Thank you very much for your professional and patient reviews.

  • It is difficult for conventional absorbers to have both wide absorbing frequency and good absorbing performance. As for the texture patterns, the structural parameters can be pre-set to achieve excellent absorption performance within a certain frequency range. Meanwhile, 3D woven fabrics have advantages of strong designability, high specific strength, high specific modulus, high damage tolerance and the ability to form complex structures at one time.
  • -10dB means that the absorption rate reaches 90%. As shown in figure in attachment can explain the real application of electromagnetic absorber at GHz range including the working frequencies of military communications and Radar waves.

Reviewer 3 Report

The authors describe the simulation and experimental validation of a model for 3D woven electromagnetic absorbers. Although the work itself seems to be valid, I am not convinced is suitable for the audience of Polymers.

Minor details:

Line 45: “absorbability”?

Figure 11: scale bars needed.

Author Response

Thank you very much for your professional and patient reviews.

We choose Polymers because we had published a review named "A Review of Electromagnetic Shielding Fabric, Wave-Absorbing Fabric and Wave-Transparent Fabric" in Polymers some time ago. As a supplement to the review mentioned above, this manuscript is suitable for publication in Polymers.

(1) Sorry for my poor English, “absorbability” has been modified into “absorption”.

(2) Scale bars had been added in Figure 11(a) and Figure 11(b).

Reviewer 4 Report

The manuscript is well elaborated and detailed in terms of depiction. In general the article can be accepted after some minor changes-

1. Figure 14, authors may introduce the error values, since they have measured 5 times for each sample.

2. The differences between theoretical and experimental reflection loss is attributed to the surface finish. Can authors introduce a physical parameter or gain factor to close the gap?

3. For all P1-P3 samples, the theoretical and experimental reflection loss values have a crossover at 17 GHz and 18 GHz. Why? Although the authors have reasoned against the same, the language is unclear.

Author Response

  1. Thank you for your professional and kind reviews. We introduced the confidence interval about Figure 14.
  2. Besides the dielectric constant of sample was different between theoretical and experimental attributed to the surface finish, the actual relative permeability of sample was not 1. In order to obtain accurate permeability, it requires complex experimental measurements of the sample, which greatly increases the complexity of experiment. In addition, with the increase of frequency, skin effect and Compton effect may occur, resulting in a larger difference between the theoretical and the experimental. Therefore, it is difficult to introduce a physical parameter or gain factor to close the gap.
  3. Because the sample itself is conductive, In the range of 16-18GHz, due to the increase of frequency, the skin effect occurs on the sample surface and the induced current was generated, so the reflection effect of the sample increases and the absorption loss decreases. However, the theoretical does not consider the skin effect, with the increase of wave frequency, the absorption loss still increases, resulting in a crossover with the experimental. A detailed explanation has been added to the manuscript.

Round 2

Reviewer 2 Report

If it will be possible to modify the textile patterns for electromagnetic wave absorption, submitted manuscript can appeal originalities.  Most important results are relation ship between the textile pattern and electromagnetic absorption properties. As a reviewer, I have to recommend that the submitted manuscript should be submitted other journal.

Author Response

  The core of this manuscript was to optimize the design of 3D wave-absorbing woven fabric by using the specified fabric structure. Because the specified fabric structure has been proved to have excellent absorbing performance, we used the specified structure to optimize the design of 3D wave-absorbing woven fabric. We choose Polymers because we had published a review named "A Review of Electromagnetic Shielding Fabric, Wave-Absorbing Fabric and Wave-Transparent Fabric" in Polymers some time ago. As a supplement to the review mentioned above, we believed that this manuscript is suitable for publication in Polymers.